# LOF variants identifying candidate genes of laterality defects patients with congenital heart disease

Sijie Liu[1][☯], Wei Wei[1][☯], Pengcheng Wang[1][☯], Chunjie Liu[1], Xuechao Jiang[2], Tingting Li[1], Fen Li[3], Yurong Wu[1], Sun Chen[1], Kun Sun[1]*, Rang Xu[2]*

1 Department of Pediatric Cardiology, Xinhua Hospital, School of Medicine, Shanghai Jiao Tong University, Shanghai, China, 2 Scientific Research Center, Xinhua Hospital, School of Medicine, Shanghai Jiao Tong University, Shanghai, China, 3 Department of Cardiology, Shanghai Children's Medical Center, School of Medicine, Shanghai Jiao Tong University, Shanghai, China

☯ These authors contributed equally to this work.
* sunkun@xinhuamed.com.cn (KS); rangxu@shsmu.edu.cn (RX)

**Data Availability Statement:** The sequencing datasets generated and analyzed during the current study are available in the China National GeneBank database (https://db.cngb.org/cnsa/). The accession number of dataset is OEP002514.

## Abstract

Defects in laterality pattern can result in abnormal positioning of the internal organs during the early stages of embryogenesis, as manifested in heterotaxy syndrome and situs inversus, while laterality defects account for 3~7% of all congenital heart defects (CHDs). However, the pathogenic mechanism underlying most laterality defects remains unknown. In this study, we recruited 70 laterality defect patients with CHDs to identify candidate disease genes by exome sequencing. We then evaluated rare, loss-of-function (LOF) variants, identifying candidates by referring to previous literature. We chose *TRIP11*, *DNHD1*, *CFAP74*, and *EGR4* as candidates from 776 LOF variants that met the initial screening criteria. After the variants-to-gene mapping, we performed function research on these candidate genes. The expression patterns and functions of these four candidate genes were studied by whole-mount in situ hybridization, gene knockdown, and gene rescue methods in zebrafish models. Among the four genes, *trip11*, *dnhd1*, and *cfap74* morphant zebrafish displayed abnormalities in both cardiac looping and expression patterns of early signaling molecules, suggesting that these genes play important roles in the establishment of laterality patterns. Furthermore, we performed immunostaining and high-speed cilia video microscopy to investigate Kupffer's vesicle organogenesis and ciliogenesis of morphant zebrafish. Impairments of Kupffer's vesicle organogenesis or ciliogenesis were found in *trip11*, *dnhd1*, and *cfap74* morphant zebrafish, which revealed the possible pathogenic mechanism of their LOF variants in laterality defects. These results highlight the importance of rare, LOF variants in identifying disease-related genes and identifying new roles for *TRIP11*, *DNHD1*, and *CFAP74* in left-right patterning. Additionally, these findings are consistent with the complex genetics of laterality defects.

(https://www.biosino.org/node/project/detail/OEP002514). The numerical data that underlie the figure and statistics are shown in S13 Table.

**Funding:** This work was supported by grants 81670210 and 81970264 from the National Natural Science Foundation of China to RX. The funders had no role in study design, data collection and analysis, decision to publish, or preparation of the manuscript.

**Competing interests:** The authors have declared that no competing interests exist.

## Author summary

Defects in laterality pattern can result in abnormal positioning of the internal organs during the early stages of embryogenesis. Patients with laterality anomalies complicated by CHD have higher mortality as compared to their CHD peers without laterality anomalies. However, the pathogenic mechanism underlying most laterality defects remains unknown. In this study, we recruited 70 laterality defects patients with CHD to identify candidate disease genes by exome sequencing. We then evaluated rare, loss-of-function variants, identifying *TRIP11*, *DNHD1*, *CFAP74*, and *EGR4* as candidates. The expression patterns and functions of these four candidate genes were studied in zebrafish models. Among the four genes, *trip11*, *dnhd1*, and *cfap74* morphant zebrafish displayed abnormalities in both cardiac looping and expression patterns of early signaling pathways. In addition, impairments of Kupffer's vesicle organogenesis or ciliogenesis were found in *trip11*, *dnhd1*, and *cfap74* morphant zebrafish. These results highlight the importance of rare, loss-of-function variants in identifying disease-related genes and display the new roles for *TRIP11*, *DNHD1*, and *CFAP74* in left-right patterning. Variants in *TRIP11* and *CFAP74* are identified in laterality defect patients for the first time.

## Introduction

Laterality defects are serious congenital malposition complexes characterized by defects of embryonic left-right (LR) patterning. They present as a range of developmental disorders, including situs inversus (SI) and heterotaxy (HTX) syndrome [1]. SI is characterized by complete, mirror-image reversal of all asymmetrical structures, whereas HTX is defined as having at least one organ discordant along the left-right axis and is traditionally classified into two groups: left atrial isomerism and right atrial isomerism [2]. Laterality defects have an estimated global prevalence of 1/10,000 and account for approximately 3~7% of all congenital heart defects (CHDs) [3,4]. Patients with laterality anomalies complicated by CHD have higher mortality as compared to their CHD peers without laterality anomalies. A previous study showed that the postoperative mortality of CHD patients with heterotaxy after surgical treatments was 4.8%, while the mortality of CHD patients without heterotaxy was 2.4% [5]. Patients with heterotaxy can have a diverse range of complex cardiac anomalies spanning all lesion types, including double outlet right ventricle (DORV), atrioventricular canal defects (AVC), anomalous pulmonary venous connection (APVC), transposition of the great arteries (TGA), single atrium (SA) and single ventricle (SV), etc. [6].

The mechanism underlying LR patterning is highly conserved among distinct classes of vertebrates. At the early somite stage, the LR axis develops symmetrically. LR asymmetry is initiated by the directional rotation of cilia in a conserved ciliated organ, the LR organizer (LRO; also referred to as the "node" in mice and the "Kupffer's vesicle" in zebrafish). Subsequently, asymmetry signals are transmitted to the left lateral plate mesoderm (LPM) to induce asymmetric expressions of genes, such as *Nodal*, left-right determination factor 2 (*Lefty2*), and paired-like homeodomain 2 (*pitx2*), and lead to LR asymmetric morphogenesis of the internal organs [2]. Several signaling pathways, such as Notch, Nodal, Hedgehog, Wnt, and transforming growth factor-beta 1 (TGF-1), which are involved in the formation of the LR axis, are conserved in all vertebrates. Previous studies have demonstrated an association between either ciliary disorders (defects of structure or function) or the dysfunction of early cell signaling during embryogenesis and laterality defects in different organisms [7–10]. In humans, mutations in the genes involved in ciliary formation (e.g., *DNAH5*, *NPHP4*) and pleiotropic signaling

pathways (e.g., *NODAL*, *CFC1*, *ACVR2B*, *LEFTY2*, *ZIC3*) have been identified in patients with laterality defects [6,11–13]. However, the known mutations only account for <20% of cases with laterality defects; [6] the etiology of laterality defects for the majority of affected patients remains unknown.

Exome sequencing (ES) is an efficient strategy to selectively sequence genomic coding regions (exons) for the identification of variations associated with human disease phenotypes, including single nucleotide variations (SNVs) and indels. ES improves the ability to detect pathologic genetic variations in complex diseases [11]. Several studies utilizing ES have demonstrated that many diseases are related to SNVs or indels, such as CHD and hypophosphatasia with mental retardation syndrome [14–16]. In previous research, both Alexander H. Li and Shuzhang Liang screened candidate laterality defect-related genes by ES [11,17]. Moreover, *MMP21* and *SHROOM3* were identified in heterotaxy patients through ES and found responsible for left-right asymmetry by further experiments in animal models [3,18]. All the studies above suggest that ES can be used for analyzing the genetic factors of laterality defects.

In our study, we identified 39 genes with multiple loss-of-function (LOF) variants through ES screening in 70 unrelated patients with laterality defects. Among these candidate variants, we found four potential genes involved in either ciliary proteome formation and function or the Nodal signaling pathway. The downregulation of three out of the four genes identified (*trip11*, *dnhd1*, and *cfap74*) in zebrafish caused disorders in both cardiac looping and the expression patterns of nodal-responsive genes (*spaw*, *lefty2*, and *pitx2*). Furthermore, our results showed that knockdown of *trip11*, *dnhd1*, and *cfap*74 in zebrafish altered Kupffer's vesicle organogenesis or ciliogenesis. To our knowledge, this is the first study to identify *TRIP11* and *CFAP74* as novel laterality defect-related genes involved in LR patterning in both humans and animals.

## Results

### Clinical data

A total of 70 unrelated Chinese patients with laterality defects were recruited for our study. All patients exhibited abnormal positioning of the internal organs and cardiac abnormalities without non-laterality-associated malformations or other syndromes. No family histories of any congenital malformations were noted in the patients' medical records. Patient ages ranged from 4 days to 16 years; 46 patients were male and 24 were female. Detailed information about the cardiac and extracardiac congenital malformations is summarized in Table 1. Total/partial anomalous pulmonary venous connection (TAPVC/PAPVC) was observed in 18 patients, double outlet right ventricle (DORV) in 28 patients, complete/partial atrioventricular canal (CAVC/PAVC) in 28 patients, and pulmonary atresia/stenosis (PA/PS) was identified in 61 patients. Thirty-one patients had malposed or transposed great arteries (MGA/TGA).

### Identification of candidate genes

To identify the genetic causes of the laterality defects, we performed ES on 70 patients and 100 healthy individuals. (Fig 1) The results revealed approximately 74,000 SNVs and 14,000 indels per individual. To identify potential disease-related genes in patients with laterality defects, variants were selected based on the following criteria: (1) located in exon or splicing region; (2) exclude synonymous variants; (3) exclude variants with allele frequency >0.1% in 1000 Genomes Project or ExAC; (4) absent in dataset of 100 healthy control individuals; (5) predicted to be disease-causing by at least one online program. According to these criteria, we identified up to 10226 potential variants, including both SNV and indel. To narrow the range

**Table 1. Cardiac and extracardiac abnormalities in the patients with laterality defects.**

| | Number of patients (%) |
|---|---|
| Sex | |
| Male | 46 (65.7%) |
| Female | 24 (34.3%) |
| Cardiac position | |
| Levocardia | 21 (30%) |
| Dextrocardia | 42 (60%) |
| Mesocardia | 6 (10%) |
| Atrial arrangement | |
| Atrial situs inversus | 23 (32.9%) |
| Isomerism of right atrial appendages | 40 (57.1%) |
| Isomerism of left atrial appendages | 5 (7.1%) |
| Ventricular arrangement | |
| Ventricular situs solitus | 21 (30%) |
| Ventricular situs inversus | 21 (30%) |
| Single ventricle | 28 (40%) |
| Bronchi | |
| Bilateral right bronchi (short) | 38 (54.3%) |
| Bilateral left bronchi (long) | 6 (8.6%) |
| Bronchial inversus | 26 (37.1%) |
| Spleen | |
| Polysplenia | 5 (7.1%) |
| Asplenia | 38 (54.3%) |
| Single right spleen | 25 (35.7%) |
| Single left spleen | 2 (2.9%) |
| Stomach | |
| Right-sided stomach | 38 (54.3%) |
| Left-sided stomach | 23 (32.9%) |
| Stomach centrally situated | 6 (8.6%) |
| Unknown | 3 (4.3%) |
| Liver | |
| Left-sided liver | 23 (32.9%) |
| Right-sided liver | 8 (11.4%) |
| Liver centrally situated | 39 (55.7%) |
| Aortic arch | |
| Left aortic arch | 32 (45.7%) |
| Right aortic arch | 38 (54.3%) |
| SVC | |
| Right SVC | 10 (14.3%) |
| Left SVC | 35 (50%) |
| Bilateral SVC | 25 (35.7%) |
| IVC | |
| Interrupted IVC, hemiazygos vein continuation | 2 (2.9%) |
| Interrupted IVC, azygos vein continuation | 5 (7.1%) |
| Relationship of IVC and descending aorta | |
| IVC right of spine and descending aorta left of spine | 5 (7.1%) |
| IVC left of spine and descending aorta right of spine | 19 (27.1%) |
| IVC and descending aorta same side | 35 (50%) |

(*Continued*)

**Table 1.** (Continued)

| | Number of patients (%) |
|---|---|
| IVC left of spine and descending aorta anterior of spine | 2 (2.9%) |
| IVC anterior of spine and descending aorta left of spine | 1 (1.4%) |
| IVC right of spine and descending aorta anterior of spine | 1 (1.4%) |

of options, we selected LOF variants consisting of frameshift, nonsense, and splice-site variants, and further screened 776 candidate variants. (S1 Table)

While 39 genes were identified more than once in the 776 selected variants (Table 2), they did not include any known LR pattern-related genes (e.g., *ZIC3*, *CFC1*, *NKX2.5*, *GDF1*, *NODAL*, *LEFTY1*, *LEFTY2*, *ACVR2B*, *DNAH5*, *DNAH11*, *DNAI1*, *FOXH1*, *CRELD1*, and *GALNT11*). We then examined the functions of the 39 candidate genes to correlate the laterality defect phenotypes to specific genes. All candidate genes were identified according to the following criteria: (1) participation in ciliary structure or function; (2) association with LR axis formation-associated signaling pathways, including Notch, Nodal, TGF-β, Hedgehog, and Wnt; (3) member of the ubiquitin ligase E3 family. We then identified 10 genes (*C11orf94*, *C16orf71*, *C2orf71*, *CABS1*, *CFAP74*, *CUL7*, *DNHD1*, *EGR4*, *FSIP2*, and *TRIP11*) related to LR patterning. Among them, *C2orf71* was excluded, because a previous study conducted a morphological examination of the *C2orf71* morphants revealed no gross defects in the body axis in zebrafish [19]. Considering 5 out of these 10 genes without homologous genes in zebrafish (Table 2), we finally choose *TRIP11*, *DNHD1*, *CFAP74*, and *EGR4* as our candidate genes (S1 and S2 Figs). There were 10 LOF variants located in 4 candidate genes, including 1 splicing, 3 stop gain, and 6 frameshift variants (Table 3), which were detected in 10 patients, respectively (Table 4).

To find out whether the patients identified as carriers of the four candidate genes had variants in other known laterality-related genes (*ZIC3*, *CFC1*, *NKX2.5*, *GDF1*, *NODAL*, *LEFTY1*, *LEFTY2*, *ACVR2B*, *DNAH5*, *DNAH11*, *DNAI1*, *FOXH1*, *CRELD1*, or *GALNT11*), We screened the coding sequences of these genes. We identified a nonsynonymous heterozygous variant (p. His4123Tyr) in *DNAH5* in one patient with a LOF variant in *CFAP74* (S2 Table).

## Expression patterns of candidate genes in zebrafish

We used zebrafish as a model organism to further analyze the biological function of the candidate genes in regulating organ laterality, as LR patterning processes are highly conserved across vertebrate species. The developmental expression patterns of the candidate genes in zebrafish were examined by whole-mount in situ hybridization at two stages: 12 hpf and 24 hpf. The dorsal forerunner cells cluster and migrate then generate Kupffer's vesicle (KV) at the tailbud by the 4- to 6-somite stages, which contributes to the proper LR asymmetric patterning in zebrafish [20]. As shown in Fig 2, trip11, dnhd1, cfap74, and egr4 exhibited nearly ubiquitous expression patterns at 12 hpf. Meanwhile, they had more localized expression patterns at 24 hpf: trip11 was localized to the pronephric duct and brain; both cfap74 and dnhd1 were expressed in the pronephric duct, brain, and tailbud; and egr4 mainly expressed in brain. The pronephric duct and neural tube are principal tissues that are involved in ciliogenesis in the early zebrafish embryo [21]. Previous studies in mice showed cilia transduced hedgehog signaling coordinates left-right patterning with heart looping and differentiation of the heart tube [22]. Based on these expression patterns, all candidate genes had a potential role in LR patterning and cardiovascular development.

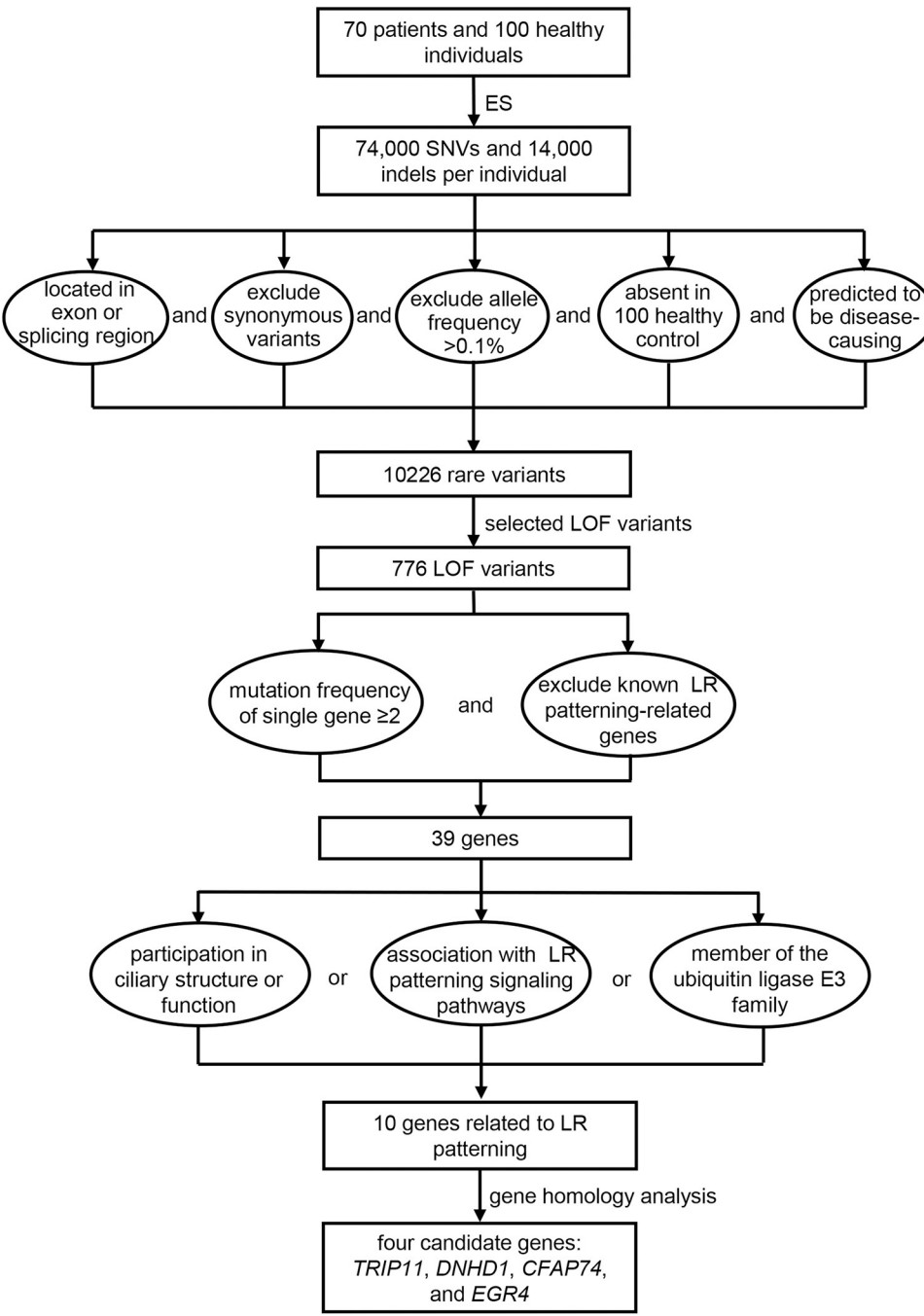

**Fig 1. Schematic of the methodology applied to identify candidate genes.** The results of ES performed on 70 patients were screened according to a series of criteria. 10226 rare variants were screened out. Among them, 776 LOF variants were selected for further analysis. 39 genes were identified from LOF variants for function and homology analysis. Finally, *TRIP11*, *DNHD1*, *CFAP74*, and *EGR4* were selected as candidate genes for further study.

## Knockdown of candidate genes disturbs cardiac looping

To test whether the candidate genes are required for LR patterning, we used MOs to knock down gene expression by disrupting mRNA splicing (*trip11* and *dnhd1*) or blocking mRNA translation (*cfap74* and *egr4*). We used *galnt11* as a positive control. In previous studies,

**Table 2. The bioinformatics information on the LOF variants of candidate genes.**

| Chromosome | Gene | OMIM number | Component | Function | Homology (Human vs zebrafish) |
|---|---|---|---|---|---|
| 5p14.1 | ACOT12 | 614315 | Acyl-CoA Thioesterase 12 | fatty Acyl-CoA Biosynthesis and Metabolism | 60.46% |
| 9p13.3 | ANKRD18B | 618930 | Ankyrin Repeat Domain 18B | nucleotide binding | 12% |
| 12q24.12 | ATXN2 | 601517 | Spinocerebellar Ataxia Type 2 Protein | negative regulator of endocytic EGFR internalization at the plasma membrane | 62.24% |
| 11p11.2 | C11orf94 | / | Chromosome 11 Open Reading Frame 94 | Cornelia De Lange Syndrome 4 With or Without Midline Brain Defects | / |
| 16p13.3 | C16orf13 | / | Chromosome 16 Open Reading Frame 1 | Left Ventricular Noncompaction | 58.78% |
| 16p13.3 | C16orf71 | / | Chromosome 16 Open Reading Frame 71 | an axonemal dynein regulator | / |
| 1q32.1 | C1orf116 | 611680 | Chromosome 1 Open Reading Frame 116 | putative androgen-specific receptor | 33% |
| 2p23.2 | C2orf71 | 613425 | Photoreceptor Cilium Actin Regulator | normal photoreceptor cell maintenance and vision | 44.58% |
| 4q13.3 | CABS1 | 618600 | Calcium Binding Protein, Spermatid Associated 1 | calcium-binding protein | / |
| 1p36.33 | **CFAP74** | / | **Cilia and Flagella Associated Protein 74** | **part of the central apparatus of the cilium axoneme** | **48.02%** |
| 19p13.3 | CFD | 134350 | Complement Factor D | functions as an adipokine and complement activation | 50.76% |
| 2p13.3 | CLEC4F | / | C-Type Lectin Domain Family 4 Member F | receptor with an affinity for galactose and fucose | 17% |
| 4p16.3 | CRIPAK | 610203 | Cysteine-rich PAK1 inhibitor | negative regulator of PAK1 | / |
| 6p21.1 | CUL7 | 609577 | Cullin 7 | an E3 ubiquitin-protein ligase complex | / |
| 11p15.4 | **DNHD1** | **617277** | **Dynein Heavy Chain Domain 1-Like Protein** | **microtubule motor activity** | **43.08%** |
| 10q26.13 | DMBT1 | 601969 | Deleted In Malignant Brain Tumors 1 | surfactant metabolism and Salivary secretion | 53.98% |
| 2p13.2 | **EGR4** | **128992** | **Early Growth Response 4** | **transcriptional regulator** | **34%** |
| 2p16.3 | FSHR | 136435 | Follicle Stimulating Hormone Receptor | G protein-coupled receptor for follitropin | 63.03% |
| 2q32.1 | FSIP2 | 615796 | Fibrous Sheath Interacting Protein 2 | play a role in spermatogenesis | / |
| 19p13.3 | FUT5 | 136835 | Fucosyltransferase 5 | fucosyltransferase activity | / |
| 14q12 | GZMH | 116831 | Granzyme H | cytotoxic chymotrypsin-like serine protease with preference for bulky | / |
| 5q35.2 | HK3 | 142570 | Hexokinase 3 | catalyzes the phosphorylation of hexose | / |
| 1q21.3 | HRNR | 616293 | Hornerin | component of the epidermal cornified cell envelopes | / |
| 22q13.33 | LMF2 | / | Lipase Maturation Factor 2 | involved in the maturation of specific proteins in the endoplasmic reticulum | 61.53% |
| 5p15.33 | LRRC14B | / | Leucine-Rich Repeat Containing 14B | a member of the PRAME family | 54.15% |
| 19p13.2 | MUC16 | 606154 | Mucin 16, Cell Surface Associated | provide a protective, lubricating barrier against particles and infectious agents at mucosal surfaces | / |
| 7q22.1 | MUC17 | 608424 | Mucin 17, Cell Surface Associated | maintaining homeostasis on mucosal surfaces | / |
| 11p11.2 | MYBPC3 | 600958 | the cardiac isoform of myosin-binding protein C | modifies the activity of actin-activated myosin ATPase | 63.89% |
| 17p11.2 | MYO15A | 602666 | Unconventional Myosin-15 | actin-based motor molecules with ATPase activity | 65.06% |
| 10q25.3 | NRAP | 602873 | Nebulin Related Anchoring Protein | anchoring the terminal actin filaments in the myofibril to the membrane and in transmitting tension from the myofibrils to the extracellular matrix | 57% |

*(Continued)*

**Table 2.** (Continued)

| Chromosome | Gene | OMIM number | Component | Function | Homology (Human vs zebrafish) |
|---|---|---|---|---|---|
| 8q21.3 | PSKH2 | / | Protein Serine Kinase H2 | transferase activity and protein tyrosine kinase activity | 46% |
| 12q24.13 | RITA1 | / | RBPJ Interacting and Tubulin Associated 1 | Tubulin-binding | / |
| 2q24.3 | SCN7A | 182392 | Sodium Voltage-Gated Channel Alpha Subunit 7 | mediates the voltage-dependent sodium ion permeability of excitable membranes | / |
| 12q14.1 | SLC16A7 | 603654 | Solute Carrier Family 16 Member 7 | symporter activity and secondary active monocarboxylate transmembrane transporter activity | 66.93% |
| 7q36.1 | SSPO | 617356 | SCO-Spondin, Pseudogene | modulation of neuronal aggregation | 52.3% |
| 5q22.1 | TMEM232 | / | Transmembrane Protein 232 | integral component of membrane | 48.42% |
| 11q14.3 | TRIM64B | / | Tripartite Motif Containing 64B | metal ion binding | 18% |
| **14q32.12** | **TRIP11** | **604505** | **Thyroid Receptor-interacting Protein 11** | **the maintenance of Golgi structure and function** | **58.42%** |
| 3q25.2 | TSEN2 | 608753 | TRNA-Splicing Endonuclease Subunit Sen2 | nucleic acid binding and tRNA-intron endonuclease activity. | 52.77% |

Bold items are candidate genes we identified.

*GALNT11*, also known as polypeptide N-acetylgalactosaminyltransferase 11, determined laterality by activating Notch signal to regulate the ratio of motile to immotile cilia at the LRO and cilia spatial distribution [23]. Meanwhile, a standard MO provided by Gene Tools was used as a negative control.

Three types of heart tube morphologies occur in zebrafish embryos: normal dextral-loop (D-loop), reversed sinistral loop (s-loop), and no loop (Fig 3A). All MOs were injected at the one-cell stage, and embryos were raised until 48 hpf, at which time the cardiac looping morphology was assessed. We found that three of the four candidate gene MOs exhibited a robust

**Table 3. The bioinformatics information on the LOF variants of candidate genes and patients.**

| ID | Gene | Mutation site | Amino acid change | Exonic Function | ExAC allele frequency | 1000G allele frequency | genomeAD pLoF (pLI) | REVEL score | CADD score |
|---|---|---|---|---|---|---|---|---|---|
| 63 | *TRIP11* | NM_004239.4:c.5855C>G | p.Ser1952* | stopgain | NA | NA | 0 | NA | 39 |
| 24 | *TRIP11* | NM_004239.4:c.4432_4433del | p.Glu1478Ilefs*8 | frameshift deletion | NA | NA | | NA | NA |
| 44 | *DNHD1* | NM_144666.3:c.2545C>T | p.Arg849* | stopgain | NA | NA | 0 | NA | 36 |
| 36 | *DNHD1* | NM_144666.3: c.11206_11207insTT | - | splicing | NA | NA | | NA | NA |
| 64 | *DNHD1* | NM_144666.3:c.7041dupC | p. Gln2348Profs*21 | frameshift insertion | NA | NA | | NA | NA |
| 60 | *CFAP74* | XM_017002642.1:c.163G>T | p.Glu55* | stopgain | NA | NA | 0 | NA | 28.3 |
| 15 | *CFAP74* | XM_017002642.1:c.1072del | p.Arg358Glyfs*52 | frameshift deletion | NA | NA | | NA | NA |
| 3 | *CFAP74* | XM_017002642.1: c.1714_1715del | p.Gly572Glnfs*35 | frameshift deletion | 0.0001 | NA | | NA | NA |
| 55 | *EGR4* | NM_001965.4:c.65dupG | p.Cys22Trpfs*7 | frameshift insertion | 0.0002 | NA | 0.01 | NA | NA |
| 72 | *EGR4* | NM_001965.4:c.65dupG | p.Cys22Trpfs*7 | frameshift insertion | 0.0002 | NA | | NA | NA |

NA, not available

**Table 4. Clinical phenotypes of laterality defects patients with LOF variations.**

| ID | Mutation site | Patients' cardiac abnormalities | Extracardiac abnormalities |
|----|---------------|--------------------------------|----------------------------|
| 63 | NM_004239.4:c.5855C>G | M, ASI, VSS, PA, PDA, ASD, VSD | BI, SRS, RS, LSL |
| 24 | NM_004239.4:c.4432_4433del | M, ASI, SV, PS, PDA, ASD | BI, SRS, RS, LSL |
| 44 | NM_144666.3:c.2545C>T | D, IRAA, SV, MGA, PS, CAVC | BRB, asplenia, LS, LCS |
| 36 | NM_144666.3:c.11206_11207insTT | D, IRAA, SV, MGA, PS, ASD | BRB, SRS, RS, LCS |
| 64 | NM_144666.3:c.7041dupC | D, ASI, VSS, TGA, PDA, ASD | BI, SRS, RS, LSL |
| 60 | XM_017002642.1:c.163G>T | D, ASI, VSI, TGA, PA, PDA, ASD, VSD | BI, SRS, RS, LSL |
| 15 | XM_017002642.1:c.1072del | D, ASI, SV, MGA, PS, PDA, CAVC | BI, SRS, RS, LSL |
| 3 | XM_017002642.1:c.1714_1715del | D, IRAA, VSI, DORV, CAVC | BRB, asplenia, LS, RSL |
| 55 | NM_001965.4:c.65dupG | D, ILAA, SV, MGA, PA, PDA | BLB, polysplenia, SCS, RSL |
| 72 | NM_001965.4:c.65dupG | L, IRAA, SV, MGA, PS, TAPVC, CAVC | BRB, asplenia, RS, LCS |

D dextrocardia, M mesocardia, ASI atrial situs inversus, IRAA isomerism of right atrial appendages, ILAA isomerism of left atrial appendages, VSS ventricular situs solitus, VSI ventricular situs inversus, SV single ventricle, PA pulmonary atresia, PS pulmonary stenosis, MGA malposed great arteries, TGA transposed great arteries, DORV double outlet right ventricle, PDA patent ductus arteriosus, CAVC complete atrioventricular canal, ASD atrial septum defect, VSD ventricle septum defect, BI bronchial inversus, BRB bilateral right bronchi (short), BLB bilateral left bronchi (long), SRS single right spleen, SLS single left spleen, RS right-sided stomach, LS left-sided stomach, SCS stomach centrally situated, LSL left-sided liver, RSL right-sided liver, LCS liver centrally situated

cardiac looping phenotype. 7.2–9.8% of the negative controls exhibited an abnormal phenotype, while 31.2% of *galnt11* morphants exhibited an abnormal phenotype ($P < 0.001$). The phenotypes of the *trip11*, *dnhd1*, *and cfap74* morphants were significantly different from that of the negative control, with 30.7–48.5% of embryos showing either an S-loop or no loop ($P < 0.001$). Yet knockdown of *egr4* had no significant effect on heart looping ($P > 0.05$) (Fig 3B). In addition, when Cas9 protein and guide RNA (gRNA; *trip11* gRNA, *dnhd1* gRNA, and *cfap74* gRNA) were co-injected into one-cell stage zebrafish embryos, both injected and F0 embryos exhibited a disrupted cardiac looping ratio ($P < 0.001$) compared to the negative control (injected with cas9 protein only; Fig 3D). However, knockout of *egr4* had no significant effect on heart looping compared with the negative control. In detail, 0–3.9% of negative controls exhibited an abnormal phenotype, while the percentages of *trip11*, *dnhd1*, *cfap74*, and *egr4* exhibiting abnormal phenotypes were 18.99–26.56%, 18.06–20.51%, 15–17.3%, and 1.59–3.75% respectively.

We then performed rescue experiments to verify whether the knockdown of each candidate gene was responsible for the abnormal phenotype. The defective LR asymmetry of the heart in the *trip11*, *dnhd1*, and *cfap74* morphants were efficiently rescued by expression of the corresponding mRNAs of the candidate genes in vitro and the detailed abnormal ratio after rescue ranged from 9.18% to 26.75% (Fig 3C; $P < 0.05$). Among them, there was no significant difference between *cfap74* morphants and std MO after rescue.

## Candidate genes exhibit global effects on early signaling pathways in LR development

Disrupted cardiac looping patterns can arise either by disturbances during early embryonic LR pattern development or by later morphogenesis of specific internal organs.[24] Clinical data from patients with selected variants indicated more than one organ malposition, suggesting that abnormal cardiac looping results from a disruption of early LR pattern development. To pinpoint the molecular cause of the LR defects, we examined the expression patterns of spaw, lefty2, and pitx2 in the morphants. These three genes are markers of LR patterning. In zebrafish embryos, Kupffer's vesicle initiates asymmetric orientation by inducing the lateral

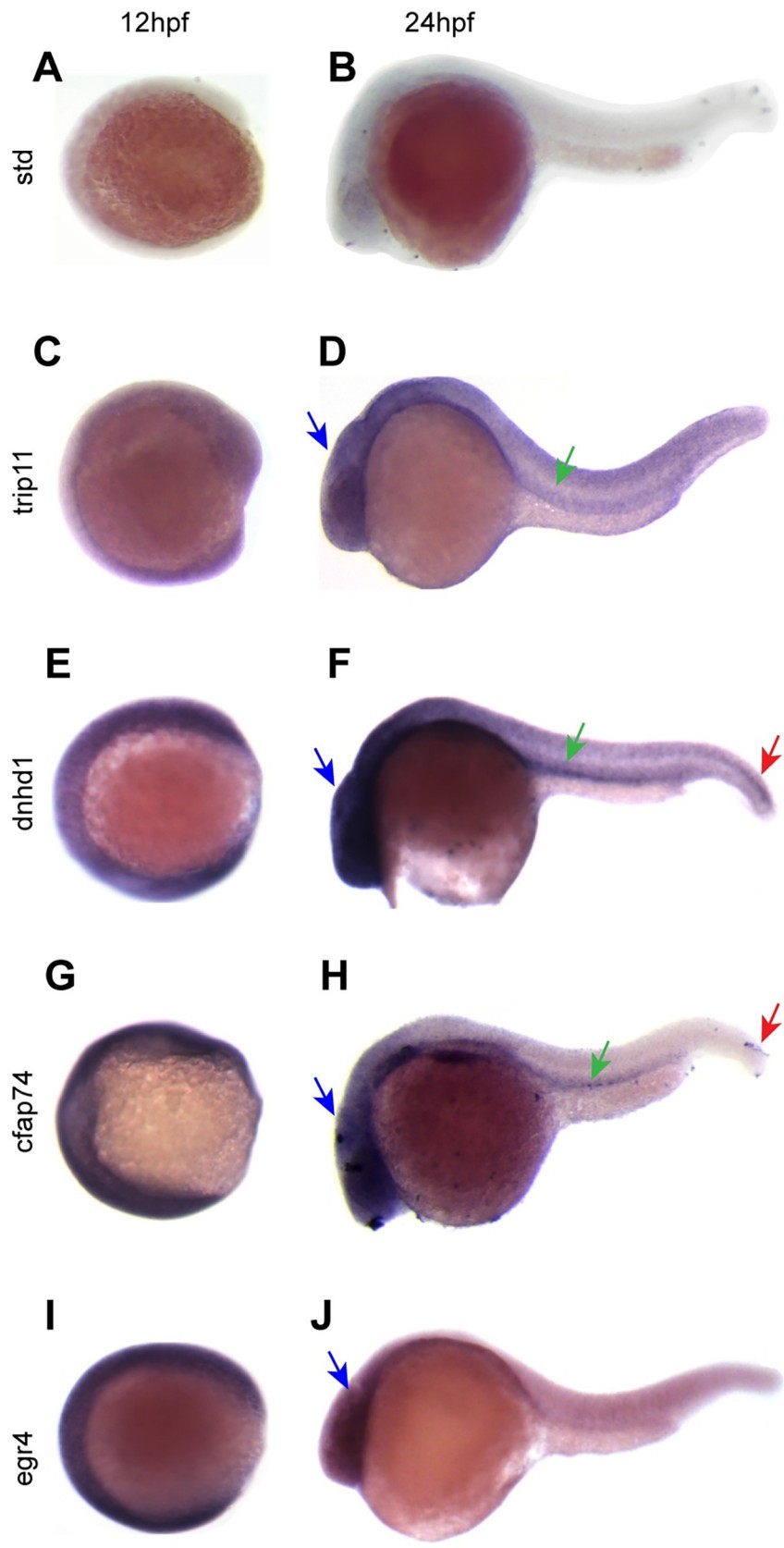

**Fig 2. Expression pattern of candidate genes at the indicated stages.** 12hpf and 24hpf. (A, C, E, G, I) Results of in situ hybridization of candidate genes and standard control at the 12hpf (8–10 somites). (B, D, F, H, J) Results of in situ hybridization of candidate genes and standard control at the 24hpf (primordium 5 stage). Lateral view of embryos with anterior to the left. pronephric duct (green arrows), tail bud (red arrow), head (blue arrows).

expression of spaw in the LPM. Then, spaw stimulates the transcription of the downstream genes *lefty2* and *pitx2*, particularly in the left side of the LPM and heart.[25] *Lefty2* encodes a nodal inhibitor belonging to the TGF-β superfamily, whereas *pitx2* encodes a transcription factor that transfers LR patterning information necessary for proper organogenesis [26].

The embryonic expression patterns of spaw, lefty2, and pitx2 exhibited either normal (left side) or abnormal (right side, bilateral, or absent) forms (Fig 4A–4C). Of the negative controls, 14.0–19.4% exhibited abnormal pitx2 expression, 18.1–23.4% displayed abnormal lefty2

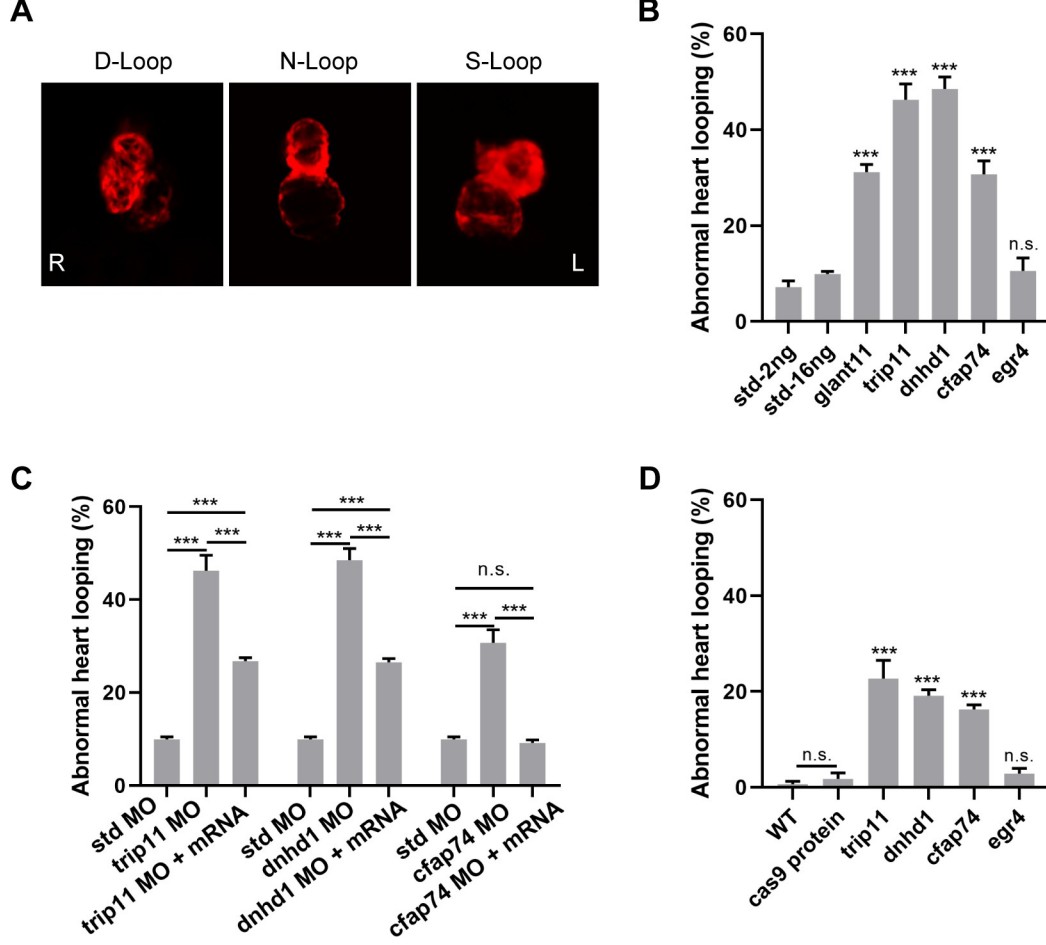

**Fig 3. Knock-down of candidate genes disturbed heart looping.** (A) Different types of zebrafish heart are shown in Tg (cmlc2: eCherry) morphants in ventral view: dextral loop (normal), sinistral loop (abnormal), and no-loop (abnormal). (B) The percentage of abnormal heart looping with morphants injected. Bars show the total percent of abnormally looped hearts including two types: no-loop and sinistral loop hearts. (C) The corresponding mRNA can rescue LR randomization caused by MO of candidate genes. (D) The ratio of the abnormal heart looping of embryos generated by co-injection of zebrafish Cas9 mRNA and gRNA. Each experiment was repeated at least 3 times with > 50 embryos examined for each group every time. Standard MO (Std) is used as the negative control. galnt11MO is used as the positive control. Chi-squared test (continuity corrected) was used in B, C and D; *P <0.05, **P < 0.01, ***P < 0.001, respectively vs. Std-16ng (*glant11*, *trip11*, *dnhd1*, and *cfap74*) or Std-2ng (*egr4*). WT, wild type.

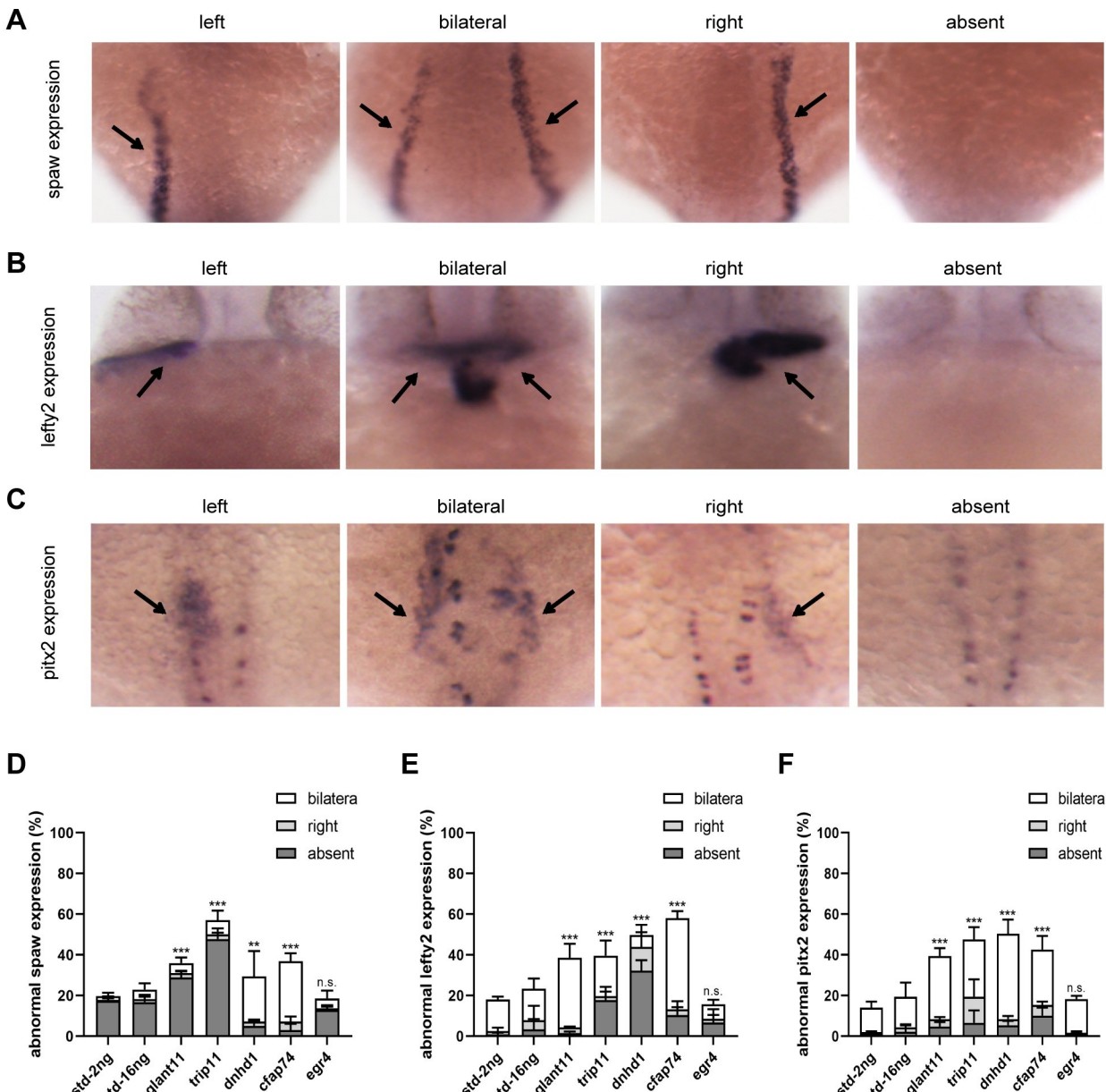

**Fig 4. The expression patterns of *spaw*, *lefty2*, and *pitx2* at 18–24 hpf.** (A, C) the expression of *spaw* and *pitx2* exhibit four patterns in LPM: left (normal), right (abnormal), bilateral (abnormal), or absent (abnormal). (B) The expression of *lefty2* shows four patterns in the cardiac field: left (normal), right (abnormal), bilateral (abnormal), or absent (abnormal). (D-F) Summary of *spaw*, *lefty2*, and *pitx2* mRNA expression in zebrafish morphants. Embryos are viewed dorsally with anterior to the top. Each experiment was repeated at least 3 times with > 50 embryos examined for each group every time. Chi-squared test (continuity corrected) was used in D, E and F; *P <0.05, **P < 0.01, ***P < 0.001, respectively vs. Std-16ng (*glant11*, *trip11*, *dnhd1*, and *cfap74*) or Std-2ng (*egr4*).

expression, and 19.8–22.9% exhibited abnormal spaw expression. Meanwhile, in morphants injected with *galnt11* MO as a positive control, 39.5% exhibited abnormal pitx2 expression, 38.6% displayed abnormal lefty2 expression, and 36.0% exhibited abnormal spaw expression (*P* < 0.001). The *trip11*, *dnhd1*, and *cfap74* morphants exhibited significant abnormal spaw, lefty2, and pitx2 expression patterns (42.6–50.4% of pitx2, 39.7–58.1% of lefty2, and 29.4–57.1% of spaw; *P* < 0.01) compared with negative control. Consistent with the phenotypic

results, *egr4* morphants exhibited no significant abnormalities in spaw, lefty2, or pitx2 expression. (Fig 4D–4F)

## The role of candidate genes in Kupffer's vesicle organogenesis and ciliogenesis

Kupffer's vesicle (KV) is a conserved ciliated epithelial structure that creates nodal flow by the directional rotation of the cilia. This flow is necessary for asymmetric gene expression [1]. Our results revealed that *trip11*, *dnhd1*, and *cfap74* might act upstream of *spaw*, impacting Nodal signaling in early development. To investigate whether the loss of these candidate genes alters LR asymmetric gene expression KV organogenesis or ciliogenesis, we first examined the morphogenesis of KV. Compared with wild-type embryos exhibiting a normal-size, rounded KV at the terminus of the notochord, *trip11*, and *cfap74* morphants displayed a smaller KV. (Fig 5A and 5B)

To explore the function of the candidate genes during ciliogenesis, the formation of KV cilia in zebrafish embryos was analyzed at the 8-somite stage. Compared with the number (average, 62±9) and length (average, 4.83±0.42 µm) of cilia in control embryos, *trip11* morphants exhibited a significant decrease in the mean number (24±10) and mean length (3.19 ±0.69 µm) of cilia ($P < 0.001$). Consistently, *dnhd1* morphants showed similar abnormalities in ciliogenesis, with an average number of 19±9 and an average length of 3.03±0.44 µm ($P < 0.001$); the average number of cilia in *cfap74* embryonic morphants was 29±12 ($P < 0.001$) and the average length was 4.14±0.77 µm ($P < 0.05$; Fig 5C–5E). Moreover, we injected arl13b-mCherry mRNA into KVs at the one-cell stage to track the movements of the cilia at the 8-somite stage. The ciliary beat frequency (CBF) in the KV of trip11 knockdown embryos was reduced significantly (Fig 5F). Dnhd1 and cfap74 knockdown embryos showed no significant difference compared with control embryos (**S1–S4 Movies**).

## Variant screening of candidate genes

We re-screened 70 patients with laterality defects for rare nonsynonymous variants of three candidate genes (*TRIP11*, *DNHD1*, and *CFAP74*) to further investigate the relationship between these genes and LR patterning. We then screened the sequences using the following criteria: (1) located in exon or splicing region; (2) exclude synonymous variants; (3) exclude variants with allele frequency >0.1% in 1000 Genomes Project or ExAC; (4) exclude variants identified in 100 normal Chinese individuals; (5) predicted to be disease-causing by at least one online program. Finally, ten rare heterozygous variants in *DNHD1* and *CFAP74* were identified S3 Table. The clinical phenotypes of these patients were shown in S4 Table.

## Discussion

Laterality defects can lead to a variety of congenital diseases, but the etiology of these defects in many patients is still unknown [27]. To explore the role of genetic variation in laterality defects, we performed ES on 70 unrelated patients and 100 healthy individuals. By analyzing the sequencing results, we identified four candidate genes. We assayed for phenocopy in a zebrafish model and then embarked on mechanistic analyses to understand the roles of these candidate genes in LR patterning and related diseases. The downregulation of these genes (*trip11*, *dnhd1*, and *cfap74*) in zebrafish resulted in disorders of both cardiac looping and the ectopic expression of nodal-responsive genes (*spaw*, *lefty2*, and *pitx2*). Meanwhile, knockdown of *trip11*, *dnhd1*, and *cfap74* altered the formation and function of cilia in KV. In addition, we identified 10 rare nonsynonymous variants in the coding sequences of *DNHD1* and *CFAP74* in patients with laterality defects.

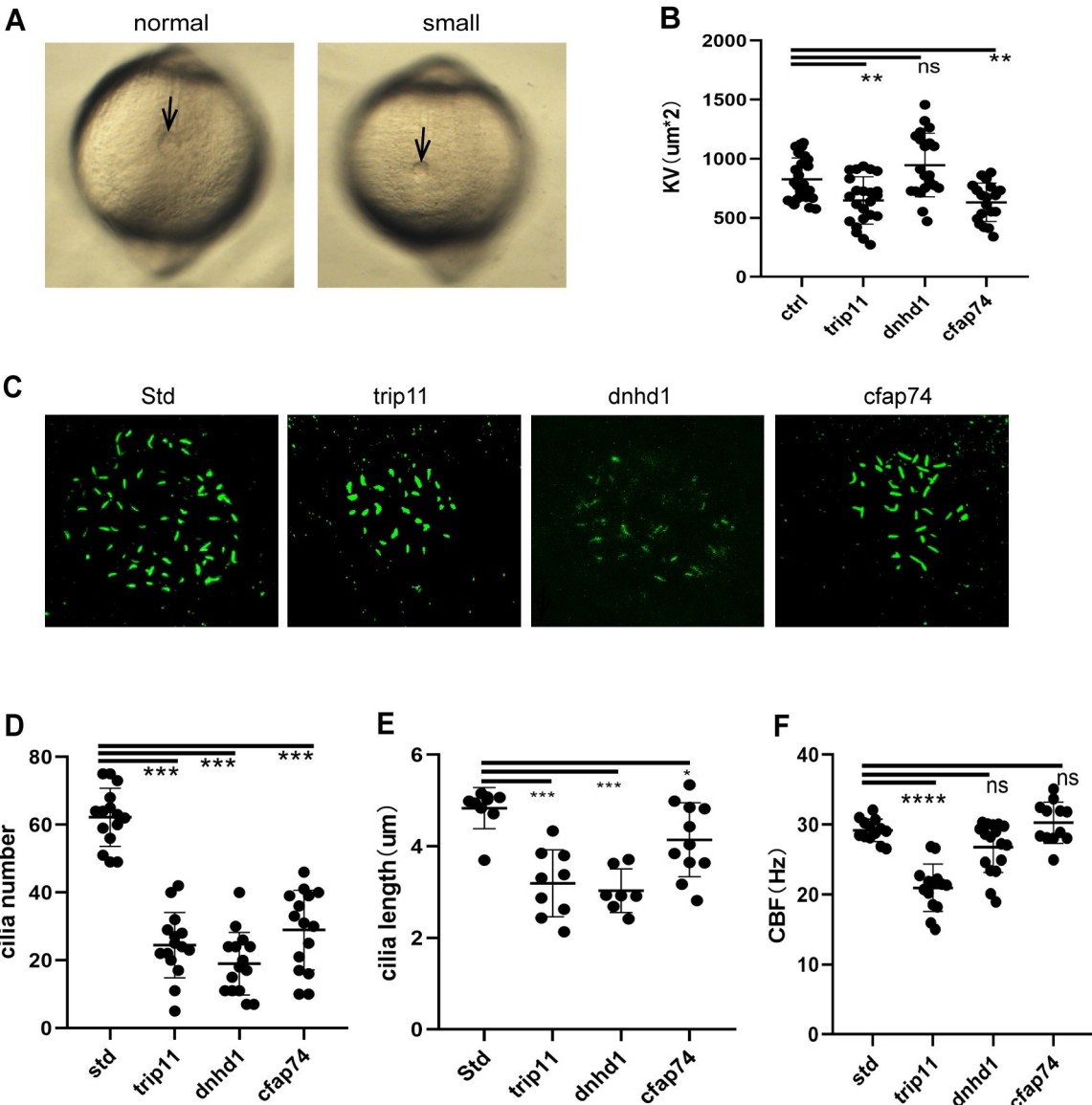

**Fig 5. Candidate genes are required for KV formation and ciliogenesis.** (A) The light-micro graph at the 8-somite stage showed different KV sizes, including normal and small. (B) The vesicle size of KV apical area. N≥19. (C) Fluorescent immunostaining of cilia in KV using anti-α-Tubulin antibodies at the 12hpf. (D) LRO Cilia number per KV was quantified and the group values were expressed as the mean ± SD. N≥15. (E) The average length of LRO Cilia per KV was quantified and the group values were expressed as the mean ± SD. N≥7. (F) The ciliary beat frequency (CBF). N≥12. Statistical significance was determined by Mann Whitney test; *P <0.05, ***P < 0.001, respectively vs. Std.

Currently, more than 100 genes have been associated with LR patterning defects in animal models, but only a few are likely candidates in humans [1]. Many studies have shown that using ES is an efficient strategy to identify pathologic variations of genes related to disease. However, since the acquisition of the patient population in this study was random, and the high number of variants identified would likely interfere with our analysis, it was critical to optimize the screening criteria for the identification of candidate genes in humans. Among all genetic variants, LOF variants generally exhibit strong pathogenicity [28]. The genetic basis of disorders such as congenital isolated hypogonadotropic hypogonadism, Aicardi-Goutiéres

syndrome, and cavernous angioma have demonstrated LOF variants that are of considerable importance in etiological research [29–31]. Moreover, previous studies of *NEK3*, *MMP21*, and *PKD1L1* have revealed that LOF variants play important roles in the pathogenesis of laterality defects in humans [3,32,33]. Thus, we screened 70 unrelated patients for rare, LOF variants that were present more than once in the same gene to narrow the scope of candidates. According to our zebrafish models, three out of four potential genes were shown to be involved in the development of asymmetry. These findings confirm that focusing on rare, LOF variants is a good approach to identify candidate genes, and the presence of more than one LOF variant in the same gene is more likely to elicit abnormal functional effects.

Our results suggest that *TRIP11*, *DNHD1*, and *CFAP74* are involved in Nodal signaling cascade by participating in KV development or ciliogenesis, which in turn contributes to LR pattern formation. These results confirm the importance of KV in LR patterning. Established genes of left–right patterning defects often involve the left-right organizer (LRO) [34]. According to the research conducted by Jason et al, the size of the KV lumen may have a significant impact on the flow dynamics necessary for LR development [35]. Besides, these results indicate the importance of ciliogenesis and ciliary function in left-right patterning. A previous study performed by Knowles et al showed that around 50% of primary ciliary dyskinesia patients have organ laterality defects [36].

The thyroid hormone receptor interactor 11 (*TRIP11*) gene encodes the Golgi microtubule-associated protein 210 (GMAP-210), which belongs to the golgin family and has been proposed to function in maintaining the morphologic and function of the Golgi apparatus [37]. Former research has shown that the loss of *TRIP11* results in developmental problems related to the defective formation and function of cilia. The lack of GMAP-210 in mice has been shown to cause lethality in neonates with diverse phenotypes, including growth restriction, tetralogy of Fallot, ventricular septal defects, and lung hypoplasia [38]. In addition, *TRIP11* male germ cell-specific conditional knockout mice exhibit infertility [39]. Heart and lung development, as well as male fertility, require normal ciliary function. Furthermore, cilia are of great importance in LR asymmetry. Our study is the first to show that *trip11* exhibited ubiquitous expression patterns at 12 hpf and localized to the pronephric duct and brain at 24 hpf in zebrafish embryos. *Trip11* knockdown led to apparent abnormalities in cardiac looping. Further, cultured mouse embryonic kidney cells obtained from GMAP-210-deficient mice displayed shortened cilia with reduced polycystin-2 levels in a previous study [38]. Interestingly, we observed similar length defects in *trip11* morphants and found that *trip11* knockdown impaired ciliary motion. It is known that GMAP-210 anchors IFT20 to the Golgi apparatus, which is required for IFT20 to sort and transport ciliary membrane proteins, such as polycystin-2 [37]. Previous work has indicated that polycystin-2 acts as a regulator of cilia length to participate in the regulation of flow-induced signaling. Polycystin-2 is also required for sensing ciliary motility in LR axis determination [40]. The involvement of both IFT20 and polycystin-2 may explain why impairments in both the structure and function of cilia were observed in *trip11* morphants.

Few studies have been performed on the function of either *DNHD1* or *CFAP74*. *DNHD1*, dynein heavy chain domain 1, encodes a ciliated structural protein in the dynein heavy chain. A homozygous missense variant of *DNHD1* was reported in patients with SI-like phenotypes (complex heart defects with incomplete intestinal rotation), but no further studies were conducted. Noteworthy, research conducted by Yue-Qiu Tan showed that *DNHD1* bi-allelic variants were identified in athenoteratozoospermia patients and these patients presented flagellar axoneme defects [41]. DNHD1 may be involved in the developmental process by participating in cilia assembly. Previous research has noted the association of several other dynein heavy chain-encoding genes, such as *DNAH1*, *DNAH5*, *DNAH9*, and *DNAH11*, with ciliary primary

dyskinesia (PCD) and laterality defects [42–45]. PCD is a serious inherited disorder that results from defects in ciliary and flagellar axoneme substructures with LR laterality developing in 50% of affected individuals [7]. Among these genes, a deficiency in *Dnah5*, a paralog of *Dnhd1*, causes a loss of outer dynein arms (ODAs) in embryonic LRO monocilia, leading to immotile cilia and impaired fluid flow in mice [13]. However, unlike *Dnah5*, loss of *dnhd1* did not elicit any impairment in ciliary motion, so we considered the loss of cilia number and length as influential factors of nodal flow. We first found that the loss of dnah1 causes disturbances in cardiac looping and exhibited global defects in early signaling pathways in zebrafish, which likely result from impaired ciliogenesis.

*CFAP74* (cilia- and flagella-associated protein 74) is a protein-coding gene that is reportedly linked to olfactory function [46]. It is highly expressed in the testes and lungs, with normal cilia playing crucial roles in both sperm motility and respiratory function.[47] Recently, biallelic mutations in *CFAP74* were identified in two patients with potential PCD and multiple morphological abnormalities of sperm flagella (MMAF), but no laterality defects were found in these patients [48]. Abnormalities in CFAP family members appear to be associated with both PCD and MMAF. For example, *CFAP53* was found in patients with laterality disorders; its deficiency in animal models was shown to result in anomalous LR asymmetry [49–51]. Further, the specific knockdown of *CFAP53* in zebrafish exhibited ultrastructural defects characterized by the severe reduction of ODAs and nonmotile cilia in KV [50,51]. Prediction of CFAP74 function based on its protein structure suggests a role in the ciliary movement, as it is part of the central apparatus of the cilia axoneme. Previous research provided indirect evidence of this function by showing that a mutation in *CFAP74* interfered with the assembly of the axoneme structure and function of the sperm flagellum [48]. Therefore, *CFAP74* loss of function was considered to disrupt the structure of cilia with a reduction in both the number and length of cilia. Similar to *DNHD1*, the loss of *CFAP74* caused the loss of cilia number and length, while did not impair ciliary motion, so we considered loss of *CFAP74* influencing ciliary function by affecting its formation. Based on these results, we suspect that *TRIP11*, *DNHD1*, and *CFAP74*, which are structurally related ciliary proteins in KV, act upstream of the Nodal signaling cascade and their decreased expression affects ciliary function.

As previously described, a single genetic mutation may not result in an obvious phenotype, but mutations in highly pleiotropic genes could have subclinical effects on phenotypes, and their cumulative effect could impact the fitness of its carrier [52]. Hence, we took the additive effects of different variants into account in our study; such considerations have also been proposed in autism spectrum disorder, mitochondrial encephalomyopathy, and several cancers [53–55]. Similar effects may exist in patients with LOF variants in *CFAP74* who also have one nonsynonymous heterozygous variant (p.His4123Tyr) in *DNAH5*. The multi-genetic basis of our results is consistent with the complexity of the development of LR asymmetry; however, to verify this hypothesis, experimental modeling and further research are needed.

Early growth response 4 (*EGR4*) is a transcriptional regulator involved in mitogenesis and differentiation [56]. In our study, knockdown of *EGR4* in zebrafish did not exhibit abnormal LR patterning.

Our research highlights the importance of rare, LOF variants in the identification of novel candidate genes in random patients with laterality defects. In addition, our functional studies illustrate those three potential genes, two of which have never been associated with LR asymmetry in either humans or animals previously, may be essential for the development of LR patterning. However, according to the guidelines, we do not have sufficient experimental and informatic support that the variants in the three genes are causal for laterality defects [57,58]. This is the limitation of our study. *TRIP11*, *DNHD1*, and *CFAP74* are more likely to be candidates with new roles in left-right patterning. Meanwhile, *TRIP11* and *CFAP74* are associated

with laterality defects for the first time. We provide preliminary evidence for their potential pathogenicity in laterality defects, and they are hoped to be further confirmed in larger groups of patients in the future. Overall, these findings have broadened our insights into the complex genetics of laterality defects and the pathogenic mechanism involved.

## Methods

### Ethics statement

This study was approved by the Local Ethics Committees of Xinhua Hospital (Shanghai, China) and Shanghai Children's Medical Center (SCMC). The serial numbers are XHEC-C-2012-018 and SCMC-201004 respectively. All parents have signed informed consent according to the guidelines of the medical ethics committee of Xin Hua Hospital and SCMC.

### Case ascertainment

Our study recruited 70 patients with laterality defects and 100 healthy individuals from Xinhua Hospital and Shanghai Children's Medical Center (SCMC). All patients included in this research were diagnosed with laterality defects and confirmed by ultrasonography, echocardiography, cardiac catheterization examinations, X-ray, computed tomography, and other operation recordings.

Patients exhibiting complex congenital heart defects and abnormal arrangement of the visceral organs were included, while those with a known chromosome abnormality or Mendelian gene syndrome and another major congenital malformation not associated with laterality defects were excluded.

### Exome sequencing and variants screening

DNA was extracted from peripheral blood samples obtained from each patient using DNeasy Blood Kit (Qiagen, Duesseldorf, Germany) according to the manufacturer's instructions. DNA of the cases and controls were sent to a commercial provider (Shanghai Biotechnology Co., Ltd., Shanghai, China), which performed sequencing services using the Illumina HiSeq2500 platform. Read mapping to hg19 was performed with Burrows-Wheeler alignment (BWA (0.7.12)). The coverage is more than 99% with a mean depth of more than 60x. The variants were annotated by ANNOVAR with a combination of databases.

Variants with more than 0.001 alternative allele frequencies in 1000 Genomes Project (http://www.1000genomes.org/) and ExAC (http://exac.broadinstitute.org), as well as those existing in control individuals were excluded. Pathogenicity of all the variants was predicted by online programs including SIFT (http://provean.jcvi.org/index.php), Polyphen-2 (http://genetics.bwh.havard.edu/pph2/), Mutation Taster (http://www.mutationtaster.org/), gnomeAD (http://gnomad-sg.org/about), REVEL (https://sites.google.com/site/revelgenomics/) and CADD (http://cadd.gs.washington.edu/score). The variants identified were confirmed by Sanger sequencing. The amplification reactions were carried out on an Applied Biosystems Veriti Cycler (Life Technologies Corporation, USA) with the following cycling program: 98˚C for 5 minutes and amplified for 35 cycles, each consisting of 30 seconds at 98˚C, 30 seconds at 55–63˚C, and 30 seconds at 72˚C per 1kb, followed by a 3-minute extension at 72˚C. The sequences of Sanger sequencing primers are provided in S5 Table.

### Zebrafish strains

Adult zebrafish (albino and AB line) and red-fluorescent labeled zebrafish (cmcl2: mcherry) were raised under standard laboratory conditions using an automatic fish housing system

(ESEN, Beijing, China) at 28°C. Wild-type embryos were obtained from adult zebrafish and raised in Holtfreter's solution at 28.5°C. All zebrafish experiments were performed at the Institute of Neuroscience, Chinese Academy of Sciences, under the guidelines of standard protocols. The stages of embryos were determined according to their developmental morphology [59].

## The whole-mount in situ hybridization

The whole-mount in situ hybridization was performed according to the previously described protocol [60]. Probes of the following genes were used in our research: *spaw*, *pitx2*, *lefty2*, *trip11*, *dnhd1*, *cfap74*, *egr4*. The anti-DIG RNA probes were synthesized with a length of 600–1300 nucleotides. Among them, the probes of *trip11*, *dnhd1*, *cfap74*, and *egr4* were synthesized by the company GENEWIZ (Suzhou, China) and then subcloned into the pGEM-T vector. The coding sequence DNA of *spaw*, *pitx2*, and *lefty2* was amplified using specific primers and then also subcloned into the pGEM-T vector. The sequences and primers are listed in S6 Table.

## Morpholino oligo injection and target gene knockdown

The standard control morpholino oligo (MO) and MOs targeting candidate genes were purchased from Gene Tools (Philomath, OR, USA). The standard control MO is a 25-mer oligo with the sequence: 5'-CCTCTTACCTCAGTTACAATTTATA-3'. According to Gene Tools' protocol, MOs were diluted to different working concentrations using nuclease-free water and were pressure-injected into one-cell-stage embryos by a Picospritzer II injector. The MOs for examining heart looping and scoring of *pitx2*, *lefty2*, and *spaw* expression in morphants were used in dosages ranging from 2 ng to 16 ng: 4ng *trip11* MO, 4ng *dnhd1* MO, 16ng *cfap74* MO, 2ng *erg4* MO, and 8ng *galnt11* MO per embryo. As negative controls, we injected 2 and 16 ng of standard control MO separately. A summary of MO doses and sequences is provided in S7 Table.

## The effectiveness evaluation of the MOs

The effectiveness evaluation methods of candidate genes are different based on different MO principles. The effectiveness evaluation method of MOs that inhibit splicing (splicing MO) targeting *trip11*/*dnhd1* is to directly detect whether the normal splicing of the original mRNA transcript has been changed by RT-PCR. The RT-PCR validation was performed according to the protocol of SYBR Premix Ex Taq II (Applied TaKaRa, Japan). The sequences of three pairs of primers are provided in S8 Table.

The MOs targeting *cfap74* and *egr4* are translation-inhibiting MO (MO-ATG). We measured the effectiveness of *egr4* MO by western blotting, while determining that of *cfap74* MO by in vitro reporter gene methods as lacking zebrafish antibodies against *cfap74*. Briefly, a pair of oligos that contain the MO target sequence of candidate genes were first annealed and then recombined into PeGFP-N1 vector which expressed the fused construct consisting of MO target sequence and the coding sequence of eGFP. 100 pg of the fusion gene vector and 16ng of the *cfap74* MO or control MO were microinjected into each zebrafish embryo at the one-cell stage. The gene which expressed fluorescent protein eGFP was observed under the fluorescence microscope. The knockdown efficiencies of these MOs are illustrated in S3 Fig.

## CRISPR/Cas9-mediated gene editing

The sequences of guide RNA were designed to introduce *trip11*, *dnhd1*, and *cfap74* gene mutation in zebrafish embryos by the clustered regularly interspaced short palindromic repeats

(CRISPR)/CRISPR-associated protein 9 (Cas9) system [61]. The sequences of *trip11*, *dnhd1*, *cfap74*, and *egr4* guide RNA (gRNA) were designed to target the sequences of mature genes and constructed by the manufacturer (XINJIA Medical, Nanjing, China). According to the previously described protocol, 600 pg zCas9 protein, and 100 pg-250 pg candidate genes, gRNA was co-injected into zebrafish embryos at the one-cell stage. The specific sequences of gRNA of candidate genes are listed in S9 Table. Then, we examined the knockout efficiency in F0 embryos by PCR and sequencing analysis. The sequences of primers are listed in S10 Table and the results are shown in S4 Fig. The knockout efficiency of *trip11*, *dnhd1*, *cfap74*, and *egr4* is 100%, 62.5%, 100%, and 100%, separately.

## mRNA synthesis and injection

The full-length coding sequence DNA of *trip11*, *dnhd1*, *cfap74*, and *egr4* were synthesized by the company GENEWIZ (Suzhou, China) and then subcloned into the pCS2+ vector. The positive clones were selected by DNA sequencing to be applied for generating full-length mRNAs. The corresponding mRNAs of candidate genes were generated by T7 or SP6 mMessage mMachine kit (Ambion, America). For the rescue experiment, the mRNA and MO of candidate genes were mixed and injected into one-cell-stage embryos. The dose of mRNAs and MOs are listed in S11 Table.

## Immunostaining and confocal microscope

Embryos were fixed in 4% paraformaldehyde in PBS overnight at 4˚C, followed by dehydration in 100% ethanol at 20˚C. Embryos were rehydrated by moving into successive dilutions of methanol in PBS and then rinsed with PBST two times every 5 minutes. Embryos were then blocked at room temperature for 2 hours in 10% heat-inactivated goat serum and then stained with the anti-α-Tubulin antibody (1:2000 T7451, Sigma) overnight at 4˚C. Samples were then washed 3 times with PBST, followed by incubation with secondary antibodies, Alexa Fluor 488 conjugated anti-mouse IgG (1:500 115-545-003, Invitrogen), overnight at 4˚C. The stained embryos were then embedded with 1.5% low melting agarose and imaged using an Olympus FV3000 confocal. We measured the cilia length by 3D tracing of cilia with imageJ. The number of KV and cilia counted were shown in S12 Table.

KV apical area was quantified to visualize the KV. Embryos were observed in a bright field using an Olympus SZX7 microscope at the 8-somite stage. A region was drawn around the KV apical perimeter and then measured using ImageJ software (NIH) to quantify the vesicle size.

## High-speed cilia video microscopy

Embryos at the eight-somite stage were mounted in 1.2% agarose with the dorsal roof of the KV facing up. Movie capture was performed at 125–250 frames per second under the OLYMPUS XLPLN25XSVMP2 25x/1.00 WD 4.00 mm objective lens on a Bruker Opterra II controlled with Prairie View Software at room temperature. CBF measurements were analyzed using ImageJ (NIH) followed by Fourier analysis in MATLAB as previously described [62].

## Statistical analysis

Cilia number and length were measured using ImageJ software. All results were expressed as the mean ± SD. Differences between control and treated groups were analyzed using the chi-squared test (continuity corrected), unpaired, two-tailed t-test, and Mann-Whitney test. Results were collected from at least 3 biologically independent replicates and considered statistically significant at $P < 0.05$ and defined $^*P < 0.05$, $^{**}P < 0.01$, $^{***}P < 0.001$.

## Supporting information

**S1 Fig. Sanger sequencing shows loss-of-function variants.** (a, b) Sanger sequencing shows frameshift or nonsense variants in *TRIP11*. (c, g, h) Sanger sequencing shows frameshift, splice-region mutant alleles or nonsense variants in *DNHD1*. (i, m, n) Sanger sequencing shows frameshift or nonsense variants in *CFAP74*. (o, s) Sanger sequencing shows frameshift variants in *EGR4*. (d, e, f, j, k, l, p, q, r, v) Sanger sequencing shows normal results that did not alter the sequences.
(TIF)

**S2 Fig. Schematic representation of the domains of four candidate genes and the position variants.** (a) the position of domains and variants in *TRIP11*. ALPS, ALPS (amphipathic lipid-packing sensor) motif; GRAB, GRAB (Grip-related Arf-binding) domain; GA1, GRAB-associated region. (b) the position of domains and variants in *DNHD1*. MTBD, microtubule-binding domain. (c) the position of domains and variants in *CFAP74*. TPH, Trichohyalin-plectin-homology domain. (d) the position of domains and variants in *EGR4*.
(TIF)

**S3 Fig. The effectiveness of the MOs.** (a-b) The RT-PCR (reverse transcription-PCR) results were conducted to analyze the efficiency of sb-MOs targeting *trip11*, and *dnhd1*. Total RNA was extracted from 2 dpf zebrafish embryos. (a) The *trip11* splice blocking morpholino (sb-MO) targets the junction of intron 1–2 and exon 2 resulting in a shorter exon 2. (b) The *dnhd1* sb-MO target the junction of exon 2, and intron 2–3 results in a shorter exon 2. (c,d) The *egr4* MO and *cfap74* MO target AUG result in lower protein expression. (c) Fluorescent immunostaining of zebrafish embryo using anti-GFP antibodies in Std embryos and *cfap74* morphants. The fusion gene vector and *cfap74* MO or control MO were microinjected at the one-cell stage. (d) Western blot revealed knockdown of protein expression in *egr4* morphants. Anti-actin was used as a loading control. Proteins were extracted from 3 dpf zebrafish embryos. Std standard control; ex, exon; in, intron.
(TIF)

**S4 Fig. Sequencing analysis of the knockout results.** Sequence analysis of *trip11*, *dnhd1*, *cfap74*, and *egr4* mutations caused by co-injection of zebrafish codon-optimized protein and corresponding gRNA. The red fonts show the target sites of gRNA, yellow fonts and blanks show mutated sequences, and the green fonts show the PAM sequences.
(TIF)

**S1 Table. The bioinformatics information of 776 candidate LOF variants.**
(XLSX)

**S2 Table. The bioinformatics information on the variants of patients with selected LOF mutations.**
(DOCX)

**S3 Table. The bioinformatics information on the variants of candidate genes.**
(DOCX)

**S4 Table. Clinical phenotypes of laterality defects patients with rare nonsynonymous variants of three candidate genes.**
(DOCX)

**S5 Table. The primers of Sanger sequencing.**
(DOCX)

**S6 Table. Antisense RNA probes conducted for whole mount in situs hybridization.**
(DOCX)

**S7 Table. MO sequences, injection doses, and total embryo numbers analyzed for heart looping and gene expression.**
(DOCX)

**S8 Table. The primers of MOs' effectiveness evaluation.**
(DOCX)

**S9 Table. The sequences and doses of gRNA.**
(DOCX)

**S10 Table. The primers of CRISPR/Cas9-mediated gene editing effectiveness evaluation.**
(DOCX)

**S11 Table. MO and mRNA injection doses, and total embryo numbers analyzed for rescue.**
(DOCX)

**S12 Table. Total number of KV and cilia for cilia length and CBF.**
(DOCX)

**S13 Table. The numerical data that underlie the figure and statistics.**
(XLSX)

**S1 Movie. The motion of motile cilia in control morphants.**
(RAR)

**S2 Movie. The motion of motile cilia in trip11 morphants.**
(RAR)

**S3 Movie. The motion of motile cilia in dnhd1 morphants.**
(RAR)

**S4 Movie. The motion of motile cilia in cfap74 morphants.**
(RAR)

## Acknowledgments

We thank all the patients for their participation in this study. We are also grateful to Dr. J. L. Du for the zebrafish platform support and critical comments on the experiments.

## Author Contributions

**Conceptualization:** Sijie Liu, Sun Chen, Kun Sun, Rang Xu.

**Formal analysis:** Sijie Liu, Wei Wei, Chunjie Liu, Xuechao Jiang, Tingting Li.

**Funding acquisition:** Rang Xu.

**Investigation:** Sijie Liu, Wei Wei, Pengcheng Wang.

**Methodology:** Sijie Liu.

**Project administration:** Kun Sun, Rang Xu.

**Resources:** Fen Li, Yurong Wu, Sun Chen, Kun Sun.

**Supervision:** Kun Sun, Rang Xu.

**Validation:** Pengcheng Wang.

**Visualization:** Wei Wei.

**Writing – original draft:** Sijie Liu, Wei Wei.

**Writing – review & editing:** Rang Xu.

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
