## [Decision Letter · Decision Letter 0]

5 Apr 2022

Dear Dr xu,

Thank you very much for submitting your Research Article entitled 'LOF variants identifying candidate genes of laterality defects patients with congenital heart disease' to PLOS Genetics.

The manuscript was fully evaluated at the editorial level and by independent peer reviewers. The reviewers appreciated the attention to an important problem, but raised some substantial concerns about the current manuscript. Based on the reviews, we will not be able to accept this version of the manuscript, but we would be willing to review a much-revised version. We cannot, of course, promise publication at that time.

If you decide to revise the manuscript for further consideration at PLOS Genetics, please aim to resubmit within the next 60 days, unless it will take extra time to address the concerns of the reviewers, in which case we would appreciate an expected resubmission date by email to plosgenetics@plos.org.

[LINK]

We are sorry that we cannot be more positive about your manuscript at this stage. Please do not hesitate to contact us if you have any concerns or questions.

Yours sincerely,

Giorgio Sirugo

Associate Editor

PLOS Genetics

Hua Tang

Section Editor: Human Variation

PLOS Genetics

Reviewer's Responses to Questions

**Comments to the Authors:**

Reviewer #1: The review was uploaded as an attachment

Reviewer #2: This manuscript uses whole-exome sequencing to identify some candidate genes that cause laterality defects in patients with congenital heart disease. The authors study the expression pattern of some of the identified genes in early zebrafish embryos and functionally validate them to find a role in the establishment of LR asymmetry in these embryos. The study is very comprehensive, is overall well executed and addresses an important topic that would interest to the broad audience of the PLOS Genetics journal. I have no major comments for authors to address prior to publication or experiments that clearly must be completed before publication, but I will include several points for them to consider below for revising the manuscript and considering additional experiments that would largely improve the manuscript.

The authors selected genes known to be involved in ciliogenesis. As cilia activity is known to be essential for LR axis establishment in zebrafish embryos, it is not surprising to find out that silencing of genes related to cilia biology/functionality causes LR defects in zebrafish. In line 185, the authors state that egr4 had no significant effect on heart looping. This reviewer wonders if the lack of laterality effect of egr4 relates to the fact that is the only gene out of the 4 genes selected that does not relate to cilia function. I wonder if other candidate genes identified by this technique but not related to cilia could have been functionally tested/validated by the authors in zebrafish. I believe that the manuscript will enormously benefit from those validations.

Minor comments:

- In figure 1, tail bud yellow arrow should be read instead of red arrow.

- In Figure 2, it is stated that 3 experiments of 50 animals each were done. Please, add error bars. Also, what Tg means?

- In Figure 3 and 4, please, add scale bars.

- In fig S2, specify the name of the domains in the figure legend.

Reviewer #3: In this article, the authors study loss-of-function (LOF) variants that may play a role in laterality defects. The authors recruited and performed whole-exome sequencing on 70 patients with laterality defects and congenital heart disease and on 100 healthy individuals, from which they identified 39 genes that satisfied specific criteria for candidate variants. From these 39 genes, the authors chose TRIP11, DNHD1, CFAP74, and EGR4 as candidate genes for their study, because they were associated with functions that could play a role in LR patterning. The authors chose zebrafish as a model to study whether the knock down of these genes affect cardiac development and LR patterning. The authors argue that four of their five candidate genes (TRIP11, DNHD1, CFAP74) resulted in cardiac looping defects, ectopic expression of nodal-responsive genes and altered formation and function of cilia, suggesting that the LOF of these genes may play a role in human laterality defects.

Although the topic of concern is of great importance and also of interest to the general public, and the data presented in the manuscript may support the claims made by the authors, I have several concerns detailed below:

Major comments:

1.- In Page 8, the authors state that their analysis filtered the candidate variants to 10226 potential variants and that, to narrow the range of options, they selected for specific LOF variants, which resulted in 827 candidate variants. However, these variants did not include any known L-R pattern-related genes, which I found striking. Could the authors comment on why they think that’s the case? Are mutations of LR pattern-related genes not common?

2.- Related to the previous question, in page 17, at the end of the results question, the authors screen the data of the 70 patients for sequences of the candidate genes TRIP11, DNHD1, and CFAP74, and they only find ten rare heterozygous variants, specifically of just DNHD1 and CFAP74. How did TRIP11 appear as a candidate before if it is not found present among the patients? I guess my question is, how is table 3 different from table S2? I am not sure if I understood the differences in data analysis between the beginning and the final screenings, but I would expect to find mutations for TRIP11, DNHD1, CFAP74, and EGR4 among the 70 patients.

3.- In Figure 1, the authors show the expression pattern of the chosen candidate genes to study the effect of their LOF. In page 14, the authors argue that “based on these expression patterns, all candidate genes had a potential role in LR patterning and cardiovascular development.” I struggle to understand this claim, as their expressions seem homogeneous at 12hpf, and their localizations at 24hpf seem to be mostly in the brain or pronephric duct. Could the authors expand on this claim?

3.A.- Is there any left-right asymmetry in the expression of these genes? Maybe a dorsal view of the expression of these genes in Figure 1 would be informative.

3.B.- Is there any expression close to the KV? For example, mmp21, a gene showed to be required for normal left-right asymmetry, and which paper is cited by the authors, shows an expression restricted to the KV at 12hpf.

4.- While images characterizing possible phenotypes observed in zebrafish are shown for reference, only the summary quantification of the phenotypes observed under each mutation are provided. While I understand space constraints in the article, I believe the raw data, or at least part of it, showing the phenotypes for control and each mutant embryo (as in Figure 4C) should be provided in the supplementary material. For example, in Figure 3 A-C, the authors show the four observed expression patterns for spaw, lefty2 and pitx2, and then, Figure 3 D-F, show the quantification of these patterns for each mutant. I think it is necessary to be able to check at least representative images for such phenotypes for each mutant.

5.- It is argued that egr4 LOF does not result in laterality defects. However, its used MO dosage is significantly lower compared to such used for the other morpholinos, could that be the reason? And related to this, I was wondering why two concentrations of standard morpholinos are used as control instead of using an uninjected condition together with a standard morpholino condition as controls.

Minor comments & Typos:

1.- Line 73: “et al.” should be changed to “etc.”

2.- In Figure 4A, the authors state that different sizes of KV are found at the 10-somite stage: normal, small and tiny. However, only two sizes (normal and small) are shown in the figure.

3.- The quantifications in Figure 4E show a significant difference between std cilia length and cfap74 cilia length, however, in Figure 4C cfap74 cilia length looks similar, if not longer, than std cilia length.

4.- Table 4 shows a mutation in patient 14, while table 3 numbers this patient as 15.

Reviewer #4: This is a manuscript describing exome sequencing results on a group of children affected with heterotaxy including complex congenital heart defects. This is a well known class of birth defects with difficult genetic contributions. Some patients have Mendelian inheritance but in many others it has not been possible to find genetic causes. This paper could contribute to the literature by identifying several new candidate genes. Strengths of the work include the use of exome sequencing to survey most of the known human coding sequence and the use of zebrafish models to evaluate a highly selected group of genes that may play a role in left right patterning. Weaknesses include the modest sample size, the lack of information about inheritance in each case, and the lack of structured summary of evidence for a causal role for each gene and variant.

Critique:

1. The methods for annotation of the gene variants are not current. The investigators should include information from gnomAD not just for variant allele frequency but also for LoF tolerance measures. REVEL scores are also readily available from public resources and should be included with the other deleteriousness metrics like the CADD scores.

2. Unfortunately parent samples are not included in the analysis and so it is not possible to evaluate for de novo occurrence. If the variants were transmitted from unaffected parents it would have also been useful to verify that they did not have milder phenotypes.

3. The investigators have not used MacArthur et al (Guidelines for investigating causality of sequence variants in human disease Nature. 2014 Apr 24;508(7497):469-76. doi: 10.1038/nature13127.) or the ClinGen gene curation framework (Am J Hum Genet. 2017 Jun 1;100(6):895-906. doi: 10.1016/j.ajhg.2017.04.015) to critically evaluate the evidence that they have accumulated in this study.

4. It would be helpful to include a decision tree or similar graphical representation of the gene filtering strategy described in the Results. It is currently hard to follow how the focus was narrowed to the four genes analyzed in the zebrafish model.

5. The cases with other deleterious variants (beginning line 240 page 17) are not clearly described. The text says four genes but only three are listed. The total number of cases where there is a candidate deleterious variant among all 70 cases is not clear. The phenotypes of these cases is not clear.

**Have all data underlying the figures and results presented in the manuscript been provided?**

Reviewer #1: **No: **No numerical data that underlie the figure and statistics have been provided

Reviewer #2: None

Reviewer #3: **No: **As included in the review: While images characterizing possible phenotypes observed in zebrafish are shown for reference, only the summary quantification of the phenotypes observed under each mutation are provided. While I understand space constraints in the article, I believe the raw data, or at least part of it, showing the phenotypes for control and each mutant embryo (as in Figure 4C) should be provided in the supplementary material. For example, in Figure 3 A-C, the authors show the four observed expression patterns for spaw, lefty2 and pitx2, and then, Figure 3 D-F, show the quantification of these patterns for each mutant. I think it is necessary to be able to check at least representative images for such phenotypes in each mutant.

Reviewer #4: Yes

PLOS authors have the option to publish the peer review history of their article (what does this mean?). If published, this will include your full peer review and any attached files.

Reviewer #1: No

Reviewer #2: No

Reviewer #3: No

Reviewer #4: No

---

## [Decision Letter · Decision Letter 1]

5 Oct 2022

Dear Dr xu,

Thank you very much for submitting your Research Article entitled 'LOF variants identifying candidate genes of laterality defects patients with congenital heart disease' to PLOS Genetics.

The manuscript was fully evaluated at the editorial level and by independent peer reviewers. The reviewers appreciated the attention to an important topic but identified some concerns that we ask you address in a revised manuscript.

We therefore ask you to modify the manuscript according to the review recommendations. Your revisions should address the specific point made by reviewer #1 on gene expression data.

Yours sincerely,

Giorgio Sirugo

Academic Editor

PLOS Genetics

Hua Tang

Section Editor

PLOS Genetics

Reviewer's Responses to Questions

**Comments to the Authors:**

Reviewer #1: Reviewer comments

The authors have considerably improved the manuscript. It is now possible to follow the choices that ended with 4 genes being selected.

I am mostly satisfied by their answers to my comments.

Major comment:

Major comment 6 on CRISPR: I understand that knockout efficiency can be assessed by sequencing, but I was asking for gene expression, which the authors already do in Fig.S3 to assess the MOs effectiveness. I would insist on seeing those results, especially since the F0 show a much-reduced effect than the MOs, they are closer to the MO+mRNA results (It would maybe make sense to put all the graphs with the same y-axis range so they can be compared properly).

If the authors did not keep any of the samples from the F0, then it would be unreasonable to redo the experiments, especially as F0 would potentially differ.

Minor comments

Line 175-176 I am not convinced by the egr4 ubiquitous expression claim at 24hpf, it seems localized mostly in the brain. Although it is consistent with the lack of effect of egr4. However, this is not central to the conclusions.

Line 147. The authors should explicitly state the lack of homolog gene in zebrafish

Figure 1. lacks a legend, most of it is self-explanatory, but it would help to guide the reader through the figure.

Figure 3. Titles over the different subpanels would clarify how each of the graphs are different. (3C) The typesetting of the MO+mRNA should be fixed. 3D should have a similar y axis as 3B and 3C to ensure proper comparison.

The multiple comparison adjustment used in the tests should be indicated either in the methods or figure legend.

Reviewer #2: The authors have addressed my concerns and I have no further comments.

Reviewer #3: I would like to thank the authors for taking the time to address my concerns. I believe the text is now more clear and the results more compelling. A last note is that the figures seem to have low resolution and are very pixelated after the revision. Hope the final figures have higher resolution.

The authors have now addressed all my concerns and I am happy to recommend this article for publication.

Reviewer #4: The authors have addressed all the comments of the reviewers and made appropriate revisions.

**Have all data underlying the figures and results presented in the manuscript been provided?**

Reviewer #1: **No: **The raw data is available, but the authors are not sharing the 827 variants of interest they have identified as they are still working on those. I think these would represent the minimal dataset required to reproduce the results.

Reviewer #2: Yes

Reviewer #3: **No: **The authors state "All the sequencing dataset files are available from the the China National GeneBank

database (https://db.cngb.org/cnsa/). Data is currently being uploaded and the code

will be added afterward.". Maybe the data will be there after publication.

Reviewer #4: Yes

PLOS authors have the option to publish the peer review history of their article (what does this mean?). If published, this will include your full peer review and any attached files.

Reviewer #1: No

Reviewer #2: No

Reviewer #3: No

Reviewer #4: No

---

## [Decision Letter · Decision Letter 2]

15 Nov 2022

Dear Dr xu,

We are pleased to inform you that your manuscript entitled "LOF variants identifying candidate genes of laterality defects patients with congenital heart disease" has been editorially accepted for publication in PLOS Genetics. Congratulations!

Yours sincerely,

Giorgio Sirugo

Academic Editor

PLOS Genetics

Hua Tang

Section Editor

PLOS Genetics

Comments from the reviewers (if applicable):

Reviewer's Responses to Questions

**Comments to the Authors:**

Reviewer #1: I am satisfied with the changes and recognize the difficulty of validating the CRISPR.

**Have all data underlying the figures and results presented in the manuscript been provided?**

Reviewer #1: Yes

PLOS authors have the option to publish the peer review history of their article (what does this mean?). If published, this will include your full peer review and any attached files.

Reviewer #1: No

**Data Deposition**

http://datadryad.org/submit?journalID=pgenetics&manu=PGENETICS-D-22-00242R2

**Press Queries**

---

## [Editor Report · Acceptance letter]

25 Nov 2022

PGENETICS-D-22-00242R2 

LOF variants identifying candidate genes of laterality defects patients with congenital heart disease 

Dear Dr xu, 

We are pleased to inform you that your manuscript entitled "LOF variants identifying candidate genes of laterality defects patients with congenital heart disease" has been formally accepted for publication in PLOS Genetics! Your manuscript is now with our production department and you will be notified of the publication date in due course.

With kind regards,

Anita Estes

PLOS Genetics

On behalf of:
